# ADAPTIVE LABEL SMOOTHING WITH SELF-KNOWLEDGE

## ABSTRACT

Overconfidence has been shown to impair generalization and calibration of a neural network. Previous studies remedy this issue by adding a regularization term to a loss function, preventing a model from making a peaked distribution. Label smoothing smoothes target labels with a pre-defined prior label distribution; as a result, a model is learned to maximize the likelihood of predicting the soft label. Nonetheless, the amount of smoothing is the same in all samples and remains fixed in training. In other words, label smoothing does not reflect the change in probability distribution mapped by a model over the course of training. To address this issue, we propose a regularization scheme that brings dynamic nature into the smoothing parameter by taking model probability distribution into account, thereby varying the parameter per instance. A model in training self-regulates the extent of smoothing on the fly during forward propagation. Furthermore, inspired by recent work in bridging label smoothing and knowledge distillation, our work utilizes self-knowledge as a prior label distribution in softening target labels, and presents theoretical support for the regularization effect by knowledge distillation and the dynamic smoothing parameter. Our regularizer is validated comprehensively on various datasets in machine translation and illustrates greater results in model performance and model calibration than when training with hard targets.

## 1 INTRODUCTION

In common practice, a neural network is trained to maximize the expected likelihood of observed targets, and the gradient with respect to the objective updates the learnable model parameters. With hard targets (one-hot encoded), the maximum objective can be approached when a model assigns a high probability mass to the corresponding target label over the output space. That is, due to the normalizing activation functions (i.e. `softmax`), a model is trained in order for logits to have a marked difference between the target logit and the other classes logits (Müller et al., 2019).

Despite its wide application and use, the maximum likelihood estimation with hard targets has been found to incur an overconfident problem: the predictive score of a model does not reflect the actual accuracy of the prediction. Consequently, this leads to degradation in model calibration (Pereyra et al., 2017), as well as in model performance (Müller et al., 2019). Additionally, this problem stands out more clearly with a limited number of samples, as a model is more prone to overfitting. To remedy such phenomenon, Szegedy et al. (2016) proposed label smoothing, in which one-hot encoded targets are replaced with smoothed targets. Label smoothing has boosted performance in computer vision (Szegedy et al., 2016), and has been highly preferred in other domains, such as Natural Language Processing (Vaswani et al., 2017; Lewis et al., 2020).

However, there are several aspects to be discussed in label smoothing. First, it comes with certain downsides, namely the static smoothing parameter. The smoothing regularizer fails to account for the change in probability mass over the course of training. Despite the fact that a model can benefit from adaptive control of the smoothing extent depending on the signs of overfitting and overconfidence, the smoothing parameter remains fixed throughout training in all instances.

Another aspect of label smoothing for consideration is its connection to knowledge distillation (Hinton et al., 2015). There have been attempts to bridge label smoothing with knowledge distillation, and the findings suggest the latter is an adaptive form of the former (Tang et al., 2021; Yuan et al.,

2020). However, the regularization effect on overconfidence by self-knowledge distillation is still poorly understood and explored.

To tackle the issues mentioned above, this work presents adaptive label smoothing with self-knowledge as a prior label distribution. Our regularizer allows a model to self-regulate the extent of smoothing based on the entropic level of model probability distribution, varying the amount per sample and per time step. Furthermore, our theoretical analysis suggests that self-knowledge distillation and the adaptive smoothing parameter have a strong regularization effect by rescaling gradients on logit space. Our work validates the efficacy of the proposed regularization method on machine translation tasks and achieves superior results in model performance and model calibration compared to other baselines.

## 2 PRELIMINARIES & RELATED WORK

### 2.1 LABEL SMOOTHING

Label smoothing (Szegedy et al., 2016) was first introduced to prevent a model from making a peaked probability distribution. Since its introduction, it has been in wide application as a means of regularization (Vaswani et al., 2017; Lewis et al., 2020). In label smoothing, one-hot encoded ground-truth label ($\boldsymbol{y}$) and a pre-defined prior label distribution ($\boldsymbol{q}$) are mixed with the weight, the smoothing parameter ($\alpha$), forming a smoothed ground-truth label. A model with label smoothing is learned to maximize the likelihood of predicting the smoothed label distribution. Specifically,

$$\mathcal{L}_{ls}(\boldsymbol{x}^{(n)}, \boldsymbol{y}^{(n)}) = -\sum_{i=1}^{|C|}(1-\alpha)y_i^{(n)}\log P_\theta(y_i|\boldsymbol{x}^{(n)}) + \alpha q_i \log P_\theta(y_i|\boldsymbol{x}^{(n)}) \quad (1)$$

$|C|$ denotes the number of classes, $(n)$ the index of a sample in a batch, and $P_\theta$ the probability distribution mapped by a model. $\alpha$ is commonly set to 0.1, and remains fixed throughout training (Vaswani et al., 2017; Lewis et al., 2020). A popular choice of $\boldsymbol{q}$ is an uniform distribution ($\boldsymbol{q} \sim U(|C|)$), while unigram distribution is another option for dealing with an imbalanced label distribution (Vaswani et al., 2017; Szegedy et al., 2016; Müller et al., 2019; Pereyra et al., 2017). The pre-defined prior label distribution remains unchanged, hence the latter cross-entropy term in Equation 1 is equivalent to minimizing the KL divergence between the model prediction and the pre-defined label distribution. In line with the idea, Pereyra et al. (2017) proposed confidence penalty (ConfPenalty) that adds negative entropy term to the loss function, thereby minimizing the KL divergence between the uniform distribution and model probability distribution. Ghoshal et al. (2021) proposed low-rank adaptive label smoothing (LORAS) that jointly learns a noise distribution for softening targets and model parameters. Li et al. (2020); Krothapalli & Abbott (2020) introduced smoothing schemes that are data-dependent.

### 2.2 KNOWLEDGE DISTILLATION

Knowledge distillation (Hinton et al., 2015) aims to transfer the *dark knowledge* of (commonly) a larger and better performing teacher model to a student model (Buciluundefined et al., 2006). The idea is to mix the ground-truth label with the model probability distribution of a teacher model, the result acting an adaptive version of label smoothing (Tang et al., 2021).

$$\mathcal{L}_{kd}(\boldsymbol{x}^{(n)}, \boldsymbol{y}^{(n)}) = -\sum_{i=1}^{|C|}(1-\alpha)y_i^{(n)}\log P_\theta(y_i|\boldsymbol{x}^{(n)}) + \alpha P_\phi(y_i|\boldsymbol{x}^{(n)}) \log P_\theta(y_i|\boldsymbol{x}^{(n)}) \quad (2)$$

$\phi$ and $\theta$ denote the parameters of a teacher model and a student model respectively. Similar to label smoothing, $\phi$ remains unchanged in training; thus a student model is learned to minimize the KL divergence between its probability distribution and that of the teacher model. When $P_\phi$ follows a uniform distribution with temperature set to 1, the loss function of knowledge distillation is identical to that of uniform label smoothing.

Training a large teacher model can be computationally expensive; for this reason, there have been attempts to replace the teacher model with the student model itself, called self-knowledge distillation (Zhang et al., 2019; Yuan et al., 2020; Kim et al., 2021; Zhang & Sabuncu, 2020). TF-KD (Yuan

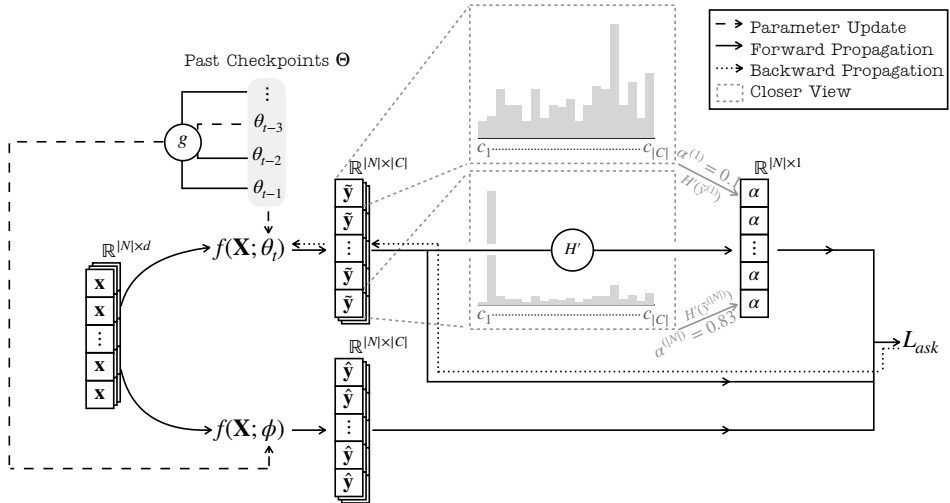

Figure 1: Overview of the proposed regularization. $d$ and $|N|$ are input dimension size and a batch size respectively. Time step is not described in the figure, yet one can easily extend the above to sequential classification tasks.

et al., 2020) trains a student with a pre-trained teacher that is identical to the student in terms of structure. SKD-PRT (Kim et al., 2021) utilizes the previous epoch checkpoint as a teacher with linear increase in $\alpha$. Zhang & Sabuncu (2020) incorporate beta distribution sampling (BETA) and self-knowledge distillation (SD), and introduce instance-specific prior label distribution. Yun et al. (2020) utilize self-knowledge distillation to minimize the predictive distribution of samples with the same class, encouraging consistent probability distribution within the same class.

## 3 METHODOLOGY

The core components of label smoothing are two-fold: smoothing parameter ($\alpha$) and prior label distribution. The components determine how much to smooth the target label using which distribution, a process that requires careful choice of selection. In this section, we illustrate how to make the smoothing parameter adaptive. We also demonstrate how our adaptive smoothing parameter and self-knowledge distillation as a prior distribution act as a form of regularization with theoretical analysis on the gradients.

### 3.1 ADAPTIVE $\alpha$

An intuitive and ideal way of softening the hard target is to bring dynamic nature into choosing $\alpha$; a sample with low entropic level in model prediction receives a high smoothing parameter to further smooth the target label. In another scenario, in which high entropy of model prediction (flat distribution) is seen, the smoothing factor is decreased.

With the intuition, our method computes the smoothing parameter on the fly during the forward propagation in training, relying on the entropic level of model probability distribution per sample, and per time step in case of sequential classification.[1]

$$H(P_\theta(\boldsymbol{y}|\boldsymbol{x}^{(n)})) = -\sum_{i=1}^{|C|} P_\theta(y_i|\boldsymbol{x}^{(n)}) \log P_\theta(y_i|\boldsymbol{x}^{(n)}) \tag{3}$$

The entropy quantifies the level of probability mass distributed across the label space; therefore, low entropy is an indication of overfitting and overconfidence (Pereyra et al., 2017; Meister et al., 2020).

Since entropy does not have a fixed range between 0 and 1, one simple scheme is to normalize the entropy with maximum entropy ($\log |C|$). Hence, the normalization is capable of handling variable

---

[1]For notational simplicity, time step is not included in the equation hereafter.

size of class set among different datasets.

$$\alpha^{(n)} = 1 - \frac{H(P_\theta(\boldsymbol{y}|\boldsymbol{x}^{(n)}))}{\log |C|} \tag{4}$$

With this mechanism, a sample with high entropy is trained with low $\alpha$, and a sample with low entropy receives high $\alpha$. The computation for $\alpha$ is excluded from the computation graph for the gradient calculation, hence, the gradient does not flow through adaptive $\alpha^{(n)}$.

There are two essential benefits of adopting the adaptive smoothing parameter. As the smoothing extent is determined by its own probability mass over the output space, the hyperparameter search on $\alpha$ is removed. Furthermore, it is strongly connected to the gradient rescaling effect on self-knowledge distillation, which will be dealt in Section 3.3 in detail.

## 3.2 Self-Knowledge As A Prior

Similar to (Kim et al., 2021; Liu et al., 2021), our regularizer loads a past student model checkpoint as teacher network parameters in the course of training. The intuition is to utilize past self-knowledge which is a generalized supervision, thereby hindering the model from overfitting to observations in the training set.

$$\phi = \underset{\theta}{\operatorname{argmax}}\, g(f(X'; \theta_i), Y'), \text{ where } \theta_i \in \Theta \tag{5}$$

$\Theta$ is a set of past model checkpoints in training, and function $f$ is a specific task, which in our work is machine translation. $X'$ and $Y'$ are sets of input and ground-truth samples from a validation dataset[2], and the function $g$ could be any proper evaluation metric for model generalization (i.e. accuracy).[3] Our work utilizes the $n$-gram matching score, BLEU (Papineni et al., 2002) being the function for finding the prior label distribution ($\boldsymbol{q} \sim P_\phi$).

Combining the adaptive smoothing parameter and self-knowledge as a prior distribution, our loss function is as follows:

$$\begin{aligned} \mathcal{L}(\boldsymbol{x}^{(n)}, \boldsymbol{y}^{(n)}) = &-\sum_{i=1}^{|C|}(1 - \alpha^{(n)})y_i^{(n)} \log P_\theta(y_i|\boldsymbol{x}^{(n)}) \\ &+ \alpha^{(n)}P_\phi(y_i|\boldsymbol{x}^{(n)}) \log P_\theta(y_i|\boldsymbol{x}^{(n)}) \end{aligned} \tag{6}$$

## 3.3 Gradient Analysis

Tang et al. (2021); Kim et al. (2021) theoretically find that the success of knowledge distillation is related to the gradient rescaling in the logit space; the difficulty of a sample determines the rescaling factor, and difficult-to-learn samples receive higher rescaling factors than those of the easy-to-learn samples.[4] We further extend the gradient analysis in the perspective of regularization effect, and discuss the importance of the adaptive smoothing parameter.

Before dissecting the gradients, we first set a hypothesis: *teacher network makes a less confident prediction than that of the student.* In self-knowledge distillation with a past checkpoint as the teacher, the assumption holds true. The expected predictive score on target label by the teacher model is lower than that of the current checkpoint *in training* (Kim et al., 2021).

The gradient with respect to the logit ($\boldsymbol{z}$) by the cross entropy loss ($\mathcal{L}_{ce}$) is as follows[5]:

$$\frac{\partial \mathcal{L}_{ce}}{\partial z_i} = P_\theta(y_i) - y_i \tag{7}$$

---

[2]Note that validation dataset is used to calculate generalization error, not for training. There is no parameter change by the function $g$.

[3]Depending on the objective of the function $g$, $\operatorname{argmin}_\theta$ can also be used.

[4]For further understanding of gradient rescaling with knowledge distillation, please refer to Proposition 2 in (Tang et al., 2021).

[5]For notational simplicity, $\boldsymbol{x}$ is omitted hereafter.

With knowledge distillation ($\mathcal{L}_{kd}$), the gradient on the logit is

$$\frac{\partial \mathcal{L}_{kd}}{\partial z_i} = (1-\alpha)(P_\theta(y_i) - y_i) + \alpha(P_\theta(y_i) - P_\phi(y_i)) \tag{8}$$

The following compares the ratio of the gradient from knowledge distillation and with that of the cross entropy.

$$\frac{\partial \mathcal{L}_{kd}/\partial z_i}{\partial \mathcal{L}_{ce}/\partial z_i} = (1-\alpha) + \alpha \frac{P_\theta(y_i) - P_\phi(y_i)}{P_\theta(y_i) - y_i} \tag{9}$$

When $i = j$, with $j$ being the index of the ground truth, it is worth noting that the denominator of the second term in Equation 9 has range $P_\theta(y_i) - 1 \in [-1, 0]$, and the range of the numerator is confined to $P_\theta(y_i) - P_\phi(y_i) \in [0, 1]$. Therefore, the equation can be written as

$$\frac{\partial \mathcal{L}_{kd}/\partial z_i}{\partial \mathcal{L}_{ce}/\partial z_i} = (1-\alpha) - \alpha \left| \frac{P_\theta(y_i) - P_\phi(y_i)}{P_\theta(y_i) - 1} \right| \tag{10}$$

The norm of the gradient drastically diminishes when there is a large difference between the predictions by the models, and when the predictive score of a student model is high, which is a sign of overconfidence. In terms of the direction of the gradient, when the following is seen,

$$(1-\alpha) < \alpha \left| \frac{P_\theta(y_i) - P_\phi(y_i)}{P_\theta(y_i) - 1} \right| \tag{11}$$

the direction of the gradients with respect to knowledge distillation becomes the opposite to that of the cross entropy, pushing parameters to lower the likelihood of the target index.

The same applies when $i$ is not equal to the target index ($i \neq j$). From Equation 9, the following can be derived.

$$\frac{\partial \mathcal{L}_{kd}/\partial z_i}{\partial \mathcal{L}_{ce}/\partial z_i} = 1 - \alpha \frac{P_\phi(y_i)}{P_\theta(y_i)} \tag{12}$$

With the generalized teacher, the expected predictive score on the incorrect labels by the teacher model is higher than that of the student model. Therefore, in addition to the shrinking norm effect, the direction of the gradient can be reversed when $1 < \alpha \frac{P_\phi(y_i)}{P_\theta(y_i)}$, similar to Equation 11; as a result, it leads to updating model parameters to increase the likelihood on the *incorrect classes*, an opposite behavior to that of the cross entropy with hard targets.

Overall, in either case, the theoretical support depicts strong regularization effects with the generalized supervision by the teacher. As label smoothing is also closely linked to the above, the theoretical support can be easily extended to label smoothing if $P_\phi(y_i)$ is replaced with $\frac{\alpha}{|C|}$ in case of uniform label smoothing, and $P(c_i)$ in unigram label smoothing.

The adaptive $\alpha$ is another factor to be discussed regarding the gradient analysis. As clearly demonstrated in Equation 11 and 12, a high $\alpha$, an indication of peak probability distribution, not only leads to drastic decrease in the gradient norm, but it is likely to make the gradient go the opposite direction to that of the cross entropy. It leads to updating the model parameters to lower the predictive score on the ground-truth target, as opposed to the effect of the cross entropy with hard targets. Furthermore, as the parameters updates are performed by aggregating the losses of samples, adaptive smoothing acts as a gradient rescaling mechanism. The gradient rescaling by adaptive $\alpha$ reweights the gradients in aggregating the losses, hence, the regularization effect is stronger on the samples with signs of overconfidence than on other samples. The use of adaptive $\alpha$ is not only intuitive in terms of tackling overconfidence, but it also serves as an important aspect in the theoretical support.

## 4 EXPERIMENT

### 4.1 DATASET & EXPERIMENT SETUP

We validate the proposed regularizer on three popular translation corpora: IWSLT14 German-English (DE-EN) (Cettolo et al., 2014), IWSLT15 English-Vietnamese (EN-VI) (Cettolo et al., 2015), and Multi30K German-English pair (Elliott et al., 2016). IWSLT14 DE-EN contains 160K sentence pairs in training, 7K in validation, and 7K in testing. IWSLT15 EN-VI has 133K, 1.5K

Table 1: The scores are reported in percentage and are averaged over three runs with different random seeds. Except for the results from the cross entropy with hard targets, denoted as Base hereinafter, other scores are absolute difference from those of Base. Bold numbers indicate the best performance among the methods.

| Corpus | Method | BLEU(↑) | METEOR(↑) | WER(↓) | ROUGE-L(↑) | NIST(↑) |
|---|---|---|---|---|---|---|
| Multi30K DE→EN | Base | 40.64 | 73.31 | 38.76 | 69.21 | 7.96 |
| | Uniform LS | +1.90 | +1.47 | -0.76 | +0.96 | +0.12 |
| | Unigram LS | +1.87 | +1.30 | -1.22 | +1.09 | +0.17 |
| | ConfPenalty | +2.50 | +1.72 | -0.81 | +1.33 | +0.15 |
| | LORAS | +1.14 | +1.00 | -0.73 | +0.63 | +0.16 |
| | TF-KD | +1.13 | +1.02 | -1.21 | +0.73 | +0.15 |
| | SKD-PRT | +1.31 | +0.95 | -0.31 | +0.54 | +0.09 |
| | BETA | +1.26 | +0.94 | -0.20 | +0.46 | +0.07 |
| | SD | +2.76 | +2.09 | -1.26 | +1.55 | +0.18 |
| | Ours | **+3.75** | **+2.91** | **-2.19** | **+2.17** | **+0.32** |
| IWSLT15 EN→VI | Base | 30.17 | 59.09 | 54.02 | 63.91 | 7.18 |
| | Uniform LS | +0.57 | +0.62 | -0.40 | +0.42 | +0.05 |
| | Unigram LS | +0.62 | +0.48 | -0.79 | +0.43 | +0.08 |
| | ConfPenalty | +0.93 | +0.56 | -1.02 | +0.59 | +0.12 |
| | LORAS | -0.04 | -0.10 | -0.23 | -0.19 | +0.02 |
| | TF-KD | -0.01 | -0.15 | -0.01 | -0.08 | -0.02 |
| | SKD-PRT | +1.03 | +0.95 | -1.31 | +0.80 | +0.16 |
| | BETA | +0.30 | +0.20 | -0.63 | +0.17 | +0.07 |
| | SD | +0.68 | +0.46 | -0.63 | +0.45 | +0.08 |
| | Ours | **+1.37** | **+1.14** | **-1.80** | **+1.05** | **+0.21** |
| IWSLT14 DE→EN | Base | 35.96 | 64.70 | 48.17 | 61.82 | 8.47 |
| | Uniform LS | +0.86 | +0.61 | -0.67 | +0.59 | +0.14 |
| | Unigram LS | +1.01 | +0.68 | -0.87 | +0.76 | +0.16 |
| | ConfPenalty | +1.15 | +0.86 | -1.08 | +0.85 | +0.19 |
| | LORAS | +0.36 | +0.23 | +0.61 | +0.16 | -0.03 |
| | TF-KD | +0.39 | +0.19 | -0.33 | +0.29 | +0.06 |
| | SKD-PRT | +1.53 | +1.11 | -1.69 | +1.25 | +0.24 |
| | BETA | +0.95 | +0.69 | -0.30 | +0.58 | +0.08 |
| | SD | +1.39 | +1.06 | -0.84 | +0.88 | +0.16 |
| | Ours | **+1.86** | **+1.55** | **-2.08** | **+1.59** | **+0.32** |

and 1.3K in training, validation, and testing dataset respectively. Lastly, 28K training, 1K validation, and 1K testing sentences are used in Multi30K dataset. Byte pair encoding (Sennrich et al., 2016) is used to process words into sub-word units.

All of the experiments are conducted with transformer architecture (Vaswani et al., 2017) on a Telsa V100. For generation, beam size is set to 4 in the inference stage. The training configuration follows the instruction of `fairseq` (Ott et al., 2019).[6] For the quality of the generated outputs, we report the popular metrics for machine translation: BLEU (Papineni et al., 2002), METEOR (Banerjee & Lavie, 2005), Word Error Rate (WER), ROUGE-L (Lin, 2004), and NIST (Doddington, 2002).

## 4.2 EXPERIMENTAL RESULT & ANALYSIS

Automatic evaluation results on the three test datasets are shown in Table 1. Though most of the methods achieve meaningful gain, the most noticeable difference is seen with our method. Our regularization scheme shows solid improvements on all of the metrics on the datasets without any additional learnable parameter. For example, the absolute gain in BLEU compared to the base method in Multi30K dataset is around 3.75, which is 9.2% relative improvement. Not only does our method excel in $n$-gram precision score, but it shows superior performance in having the longest

---
[6] https://github.com/pytorch/fairseq/tree/main/examples/translation

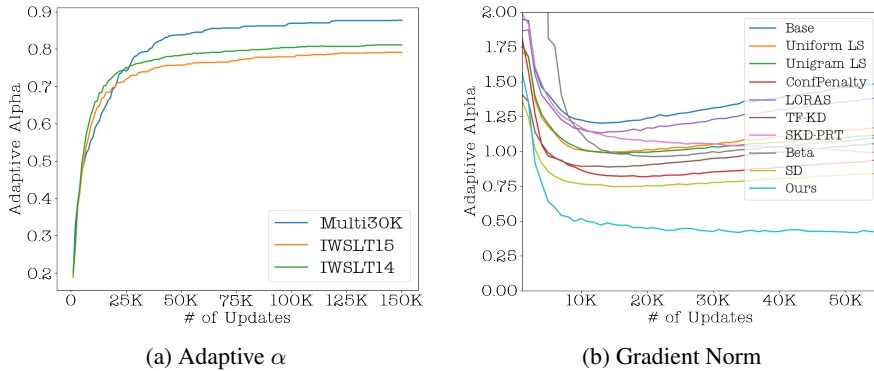

(a) Adaptive $\alpha$          (b) Gradient Norm

Figure 2: (a) displays the change in the smoothing parameter $\alpha$ throughout the training on the tested corpora. (b) plots the gradient norm of the methods on IWSLT15 EN→VI.

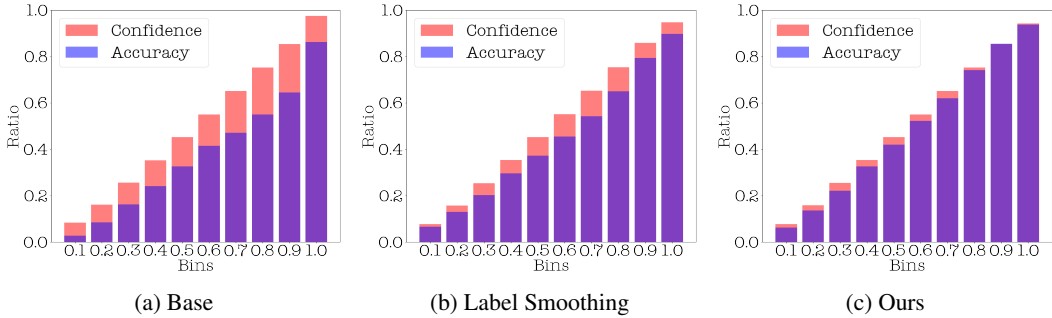

(a) Base         (b) Label Smoothing         (c) Ours

Figure 3: Reliability diagram of Base method, label smoothing, and ours. Predictions on IWSLT14 DE→EN test set are binned to 10 groups based on the predictive scores. Each bar indicates the average confidence score and accuracy of each bin.

common subsequence with the reference text, as well as in the informativeness of the $n$-grams. The empirical result demonstrates that our regularizer improves the base method across all the metrics by a large margin.

In Figure 2a, the changes in $\alpha$ during training are visualized. As expected, the smoothing parameters start with a very small number, as the entropic level must be high due to the under-fitted models. As training continues, the predictive scores of the models increase, and accordingly, adaptive $\alpha$ increases to prevent overconfidence. One notable aspect is the convergence at a certain level. Each training of the corpora ends up with a different $\alpha$, and the model in training self-regulates the amount of smoothing and the value converges.

Furthermore, our adaptive $\alpha$ affects the norm of the gradients as depicted in Figure 2b. The gradient norm of our regularizer is considerably smaller than that of the other methods. This empirical finding mainly conforms with the gradient analysis in Section 3.3, where the importance of adaptive $\alpha$ and generalized teacher model was discussed.

### 4.2.1 MODEL CALIBRATION

In addition to the automatic evaluation, in which the improved generalization is seen through the performance gains, we look into the calibrations of the models trained with the methods. Figure 3 depicts how the cross entropy with hard targets tends to make a model overconfident in prediction. In the reliability diagram, the confidence score in each bin is larger than that of accuracy, the gap of which is noticeable. Label smoothing mitigates the problem to some extent, yet the gap between the accuracy and the confidence score still remains clear. On the other hand, the proposed regularizer significantly reduces the gap, showing the enhanced model calibration.

Table 2: We report Expected Calibration Error (**ECE**) and Maximum Calibration Error (**MCE**), in percentage, on the test sets of the corpora for evaluating the calibration ability.

| Method | Multi30K DE→EN | | IWSLT15 EN→VI | | IWSLT14 DE→EN | |
|---|---|---|---|---|---|---|
| | **ECE** ($\downarrow$) | **MCE** ($\downarrow$) | **ECE** ($\downarrow$) | **MCE** ($\downarrow$) | **ECE** ($\downarrow$) | **MCE** ($\downarrow$) |
| Base | 14.95 | 26.01 | 14.05 | 20.38 | 12.98 | 19.29 |
| Uniform LS | 9.17 | 17.22 | 8.53 | 12.13 | 6.43 | 9.98 |
| Unigram LS | 9.12 | 17.78 | 7.89 | 11.71 | 6.12 | 9.46 |
| ConfPenalty | 48.21 | 73.46 | 43.94 | 59.28 | 48.19 | 57.58 |
| LORAS | 20.27 | 40.86 | 12.41 | 19.15 | 10.54 | 15.29 |
| TF-KD | 21.18 | 42.87 | 13.30 | 19.29 | 12.20 | 17.60 |
| SKD-PRT | 14.75 | 26.69 | 9.34 | 14.18 | 5.63 | 8.88 |
| BETA | 11.71 | 21.90 | 9.57 | 14.97 | 8.63 | 13.21 |
| SD | 6.87 | **12.38** | 5.01 | 9.64 | 7.82 | 13.71 |
| Ours | **4.76** | 12.41 | **2.15** | **4.40** | **1.76** | **3.64** |

The improvement in calibration is more clear with expected calibration error (ECE) and maximum calibration error (MCE) reported in Table 2. For instance, on the IWSLT14 dataset, the errors with label smoothing drop significantly in both metrics, which is around 6% absolute decrease in ECE and 10% in MCE. Nonetheless, ECE of the proposed method results in 1.76% which is around 11% absolute decrease and 86% relative improvement. In addition, our method achieves 3.64% in MCE, which is 81% relative improvement over the base method. The improved calibration with our method is seen across the datasets, demonstrating the effectiveness of our method in model calibration.

One important finding is the gap between the performance in automatic evaluation and the calibration error. Confidence penalty (Pereyra et al., 2017) is highly competitive in $n$-gram matching scores (BLEU) on all of the dataset tested. Nevertheless, the calibration error is the highest among the methods. Similar to the finding in (Guo et al., 2017), the discrepancy between the performance and model calibration exists, and it calls for caution in training a neural network when considering model calibration.

## 4.3 ABLATION STUDY

Table 3 shows the change in performance when adding our core components to the base method on the IWSLT14 dataset. When using the fixed smoothing parameter with self-knowledge as a prior, the BLEU score increases by a small margin, and the ECE does not drop significantly. In another case where the smoothing parameter is adaptive, and the prior label distribution is set to uniform distribution, there is a meaningful increase in BLEU score. However, it impairs the ECE score noticeably. We empirically find that the result mainly comes from *underconfidence* of a model. The confidence score is largely lower than that of the accuracy. In an experiment with linearly increasing $\alpha$ with self-knowledge prior, the BLEU score improves by around 1.6 score, yet the ECE score still has room for improvement. Since $\alpha$ value is shared among samples in this ablation experiment, there is no gradient rescaling by adaptive $\alpha$ which may explain ECE score being high compared to that of our adaptive $\alpha$. We

Table 3: (+) denotes adding the following components to the base method. $\alpha^{(n)}$ denotes our adaptive $\alpha$, and $\alpha^{\uparrow}$ indicates a linear increase in $\alpha$ in the course of training. SK and Uniform denote Self-Knolwedge and Uniform distribution as a prior label distribution respectively. $g_{\text{NLL}}$ and $g_{\text{BLEU}}$ indicates $g$ function set to negative log likelihood and BLEU respectively.

| Method | **BLEU** | **ECE** |
|---|---|---|
| Base | 35.96 | 12.98 |
| (+) Fixed $\alpha$ & SK | 36.27 | 13.56 |
| (+) $\alpha^{(n)}$ & Uniform | 37.30 | 18.76 |
| (+) $\alpha^{\uparrow}$ & SK | 37.52 | 5.58 |
| Ours ($g_{\text{NLL}}$) | 37.74 | 1.30 |
| Ours ($g_{\text{BLEU}}$) | 37.82 | 1.76 |

also look into a case with a different $g$ function: BLEU and Negative Log Likelihood (NLL). We observe that both $g_{\text{BLEU}}$ and $g_{\text{NLL}}$ greatly improve the scores. As $g$ has the purpose of selecting a generalized teacher from the set of past checkpoints, a proper metric would serve the purpose. In

conclusion, while the adaptive smoothing parameter plays an important role in regularizing a model, both adaptive smoothing parameter and the choice of prior label distribution greatly affect model calibration.

## 5 CONCLUSION & FUTURE WORK

In this work, we propose a regularization scheme that dynamically smooths the target label with self-knowledge. Our regularizer self-regulates the amount of smoothing with respect to the entropic level of the model probability distribution, making the smoothing parameter dynamic per sample, and per time step. The given idea is theoretically supported by gradient rescaling and direction, and the finding is backed up by the empirical results, both in model performance and calibration.

The future work is expected to validate the proposed regularization on other domains outside of Natural Language Processing. The scope of application is not confined to natural language generation tasks, and it can be easily extended to other classification tasks.

## REPRODUCIBILITY STATEMENT

For reproducibility, we report the three random seeds tested: {`0000, 3333, 5555`}. For all of the experiments, this work utilizes the transformer architecture (Vaswani et al., 2017). Both the encoder and the decoder are composed of 6 transformer layers with 4 attention heads. The hidden dimension size of the both is 512. For training configuration, the maximum tokens in a batch is set to 4,096. For optimization, Adam (Kingma & Ba, 2015) is used with beta 1 and beta 2 set to 0.9 and 0.98 respectively. We slowly increase the learning rate up to 0.005 throughout the first 4,000 steps, and the learning rate decreases from then on. The source code and training script are included in the supplementary materials.

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

## A  APPENDIX

### A.1  BASELINES

#### A.1.1  CONFIDENCE PENALTY

Confidence penalty (Pereyra et al., 2017) adds a negative entropy term to the loss function, hence the model is encouraged to maintain entropy at certain level.

$$\mathcal{L}_{cf}(\boldsymbol{x}^{(n)}, \boldsymbol{y}^{(n)}) = -\sum_{i=1}^{|C|} y_i^{(n)} \log P_\theta(y_i|\boldsymbol{x}^{(n)}) - \beta H(P_\theta(\boldsymbol{y}|\boldsymbol{x}^{(n)})) \tag{13}$$

For the regularization-specific hyperparameter, following (Meister et al., 2020), $\beta$ was set to 0.78.

#### A.1.2  TF-KD

TF-KD (Yuan et al., 2020), similar to conventional knowledge distillation, trains a teacher model prior to training a student; but it is different in that the model architecture is same with that of the student. For the hyperparameters used in this paper, we empirically find that high smoothing parameter leads to better performance. Thus, we set the smoothing parameter to 0.9 and temperature scaling to 20.

### A.1.3  SKD-PRT

SKD-PRT (Kim et al., 2021) is a self-knowledge distillation method, where a student model (epoch $t$) is trained with its own last epoch checkpoint (epoch $t-1$) in the course of training. Though the idea is similar to ours, yet there are two core differences. The first is that we find the teacher model that generalizes well with a function $g$. Another difference is that SKD-PRT linearly increases $\alpha$ throughout the training, and this practice inevitably adds two hyperparameters (`max` $\alpha$ and `max` epoch). Following the original work (Kim et al., 2021), we set that maximum $\alpha$ to 0.7 and maximum epoch to 150 in our experiments.

### A.1.4  LORAS

LORAS (Ghoshal et al., 2021) jointly learns a soft target and model parameters in training in the aim of increasing model performance and model calibration, with low rank assumption. For hyperparameters, $\eta$, $\alpha$, rank and dropout probability are set to 0.1, 0.2, 25 and 0.5 respectively.

### A.1.5  BETA & SD

Zhang & Sabuncu (2020) propose amortized MAP interpretation of teacher-student training, and introduce Beta smoothing which is an instance-specific smoothing technique that is based on the prediction by a teacher network. For SD-specific hyperparameters, this work sets $\alpha$ to 0.3 and temperature to 4.0. For BETA-specific hyperparameters, $\alpha$ and $a$ are set to 0.4, 4.0 respectively.

