# OpenReview forum: "Adaptive Label Smoothing with Self-Knowledge"
_ICLR.cc/2022/Conference — ICLR 2022 Submitted_

### Official Review · Reviewer_PF2S · 2021-10-22

**Correctness:** 4
**Technical Novelty And Significance:** 3
**Empirical Novelty And Significance:** 3
**Recommendation:** 6
**Confidence:** 3

**Main Review:**

Positive points:

+ the paper proposes a simple way of adapting the smoothing parameter alpha. And it works well in the experiments.
+ Instead of a simple distribution that is often used for smoothing, the paper proposes to use the distribution of the same model but from an earlier training epoch.
+ experimental results show improved accuracy and improved calibration compared to baselines

Negative points:

- it is not clear to me how important it is in the proposed approach that function g in  Eq 5 is the metric that is eventually used in evaluation (e.g., BLEU), rather than the training-objective itself, e.g., likelihood/cross entropy during training. Given that alpha is tuned dynamically based on function g, it seems that this could be a unique/unfair advantage of the proposed approach over the other baselines, which are only trained based on likelihood/cross-entropy, but have no access to the evolution metric (e.g., BLEU), which is different. For this reason, it could also be interesting to have experiments where the evaluation metric and training loss are the same function, e.g., log likelihood. Also, in Table 1, where different evaluation metrics are shown (BLEU, METEOR,...), is the 'Ours"-model always trained with function g = BLEU, or is function g chosen to be identical to the evaluation metric, e.g., when evaluated w.r.t. METEOR, then g-METEOR is used during training?

- also, it might be good to add the original paper that proposed knowledge distillation:  Bucilua, Caruana and Niculescu-Mizil: Model compression. KDD 2006.


**Summary Of The Paper:**

The paper proposes a simple yet effective way of smoothing the labels for each data point. The smoothing parameter alpha is dynamically adapted for each data point based on the (normalized) entropy of the predicted label-distribution for this data point. The distribution that is used for label-smoothing originates from the same but at earlier epoch of the training, i.e., a teacher-student learning framework like in knowledge distillation is employed, where the teacher is a model learned in an earlier epoch. From among these earlier model-candidates, the one is chosen that obtains the best evaluation metric g (Eq. 5). Function g does not have to be the training loss, but can be the (possibly different) evaluation metric. The validity of this approach is theoretically supported by a gradient analysis, and experimentally corroborated by improved evaluation metrics, improved calibration, and an ablation study.


**Summary Of The Review:**

A theoretically simple yet empirically effective approach to label smoothing.

+++++++++++++++++

Update based on authors' response:
Thanks for the clarifications. I will maintain my current score.

---

> ### Author Response · Authors · 2021-11-15
> **Response to Reviewer PF2S**
>
> ### Response 1
> > Given that alpha is tuned dynamically based on function g, it seems that this could be a unique/unfair advantage of the proposed approach over the other baselines, which are only trained based on likelihood/cross-entropy, but have no access to the evolution metric (e.g., BLEU), which is different.
>
> Sorry for any confusion our work may have caused. Our adaptive alpha is not tuned dynamically based on function $g$. Our adaptive alpha is computed on the fly during the forward propagation at each iteration based on the probability distribution of each time step and each sample by a model in training (Eq. 4 in the paper). Function $g$ serves as identifying a generalized teacher network from the set of the past checkpoints, and there is no update on the weight of the past checkpoint by the function $g$. Therefore, it is fair to other baselines without the function $g$. Lastly, our function $g$ does not make use of any test dataset during training, and all the scores reported in the paper are results with the test dataset. Therefore, the proposed method is trained and evaluated in a fair setting.
>
> ### Response 2
> > it could also be interesting to have experiments where the evaluation metric and training loss are the same function, e.g., log likelihood.
>
> Thank you for your insightful recommendation. As you have asked, we have tested $g$ function being the negative log likelihood. Here are the results.
>
> |Method|BLEU (&#8593;)|METEOR (&#8593;)|WER (&#8595;)| ROUGE-L (&#8593;)|NIST (&#8593;)|ECE (&#8595;)|MCE (&#8595;)|
> |:--------:|:-------------:|:------:|:------:|:------:|:------:|:------:|:------:|
> | $g$ = BLEU | 37.82|66.24|46.09	|63.41|8.79 |	1.76|	3.64|
> | $g$ = NLL | 37.74 | 66.09 | 46.04 | 63.36 | 8.76 | 1.30 | 3.18 |
>
> As depicted from the result, a proper evaluation metric would suit the purpose of function $g$. Function $g$ is present to identify a generalized teacher network from the set of past checkpoints, hence a proper evaluation metric would fulfill the purpose. We have added the result to the ablation table in the updated version.
>
> ### Response 3
> > is the 'Ours"-model always trained with function g = BLEU, or is function g chosen to be identical to the evaluation metric, e.g., when evaluated w.r.t. METEOR, then g-METEOR is used during training?
>
> Yes. In Table 1, all the scores are from a model trained with BLEU being the function $g$. Now, the ablation table contains results from a model with NLL being the function $g$.
>
> ### Response 4
> > also, it might be good to add the original paper that proposed knowledge distillation: Bucilua, Caruana and Niculescu-Mizil: Model compression. KDD 2006.
>
> Thank you for your suggestion, we have cited the paper to the modified version.

---

> ### Author Response · Authors · 2021-11-20
> **Dear Reviewer PF2S**
>
> We hoped our response helped resolve your concerns regarding the function $g$ in our work. If you have any further questions, please feel free to ask.

---

### Official Review · Reviewer_xcg8 · 2021-10-29

**Correctness:** 3
**Technical Novelty And Significance:** 3
**Empirical Novelty And Significance:** 3
**Recommendation:** 6
**Confidence:** 4

**Main Review:**

Strengths:
1.	This paper proposes the adaptive \alpha computed by the entropic level of model probability distribution per sample, which leads to updating the model parameters to lower the predictive score on the ground-truth target, as opposed to the effect of the cross-entropy with hard targets.  The hyperparameter search on \alpha in label smoothing is removed.
2.	This paper set an assumption that the teacher network makes a less confident prediction than that of the student and extends gradient analysis in the perspective of regularization effect in the proposed adaptive label smoothing.

Weakness:
1.	This paper demonstrates why self-knowledge distillation as a prior distribution is a form of regularization with theoretical analysis on the gradients. However, there is no theoretical result about the effectiveness of assigning the self-knowledge distillation to label smoothing.
2.	The authors claim that “There are a number of benefits of adopting the adaptive smoothing parameter”. However, they only show that the hyperparameter search on \alpha is removed and the adaptive smoothing parameter can be connected to the gradient rescaling effect on self-distillation. More details should be reported to show the benefits of adopting the adaptive smoothing parameter.
3.	In the ablation study, the authors only consider the fixed \alpha as the base. However, \alpha could also be changed in the training process. Therefore, the dynamic \alpha by hyperparameter searching should also be added as a base.


[1]  Learning Better Structured Representations Using Low-rank Adaptive Label Smoothing, ICLR 2020.
[2]  Adaptive label smoothing,  arXiv 2020.


**Summary Of The Paper:**

This paper focuses on adaptive label smoothing that could reflect the change in probability distribution mapped by a model over the course of training. To deal with this issue, this paper proposes a label soothing scheme that brings dynamic nature into the smoothing parameter and the prior label distribution from the distilled knowledge. Specifically, the smoothing parameter is computed with the entropic level of model probability distribution per sample on the fly during the forward propagation in training.  Besides, the prior label distribution is selected from the self-knowledge distillation. Experiments on various datasets demonstrate that the proposed adaptive label smoothing achieves state-of-the-art performance.

**Summary Of The Review:**

This paper proposes a label soothing scheme that brings dynamic nature into the smoothing parameter and the prior label distribution from the distilled knowledge. More theoretical results and empirical results should be reported to validate the effectiveness of the proposed adaptive label smoothing. The contributions are significant and somewhat new. Aspects of the contributions exist in prior work.

---

> ### Author Response · Authors · 2021-11-15
> **Response to Reviewer xcg8**
>
> ### Response 1
> > This paper demonstrates why self-knowledge distillation as a prior distribution is a form of regularization with theoretical analysis on the gradients. However, there is no theoretical result about the effectiveness of assigning the self-knowledge distillation to label smoothing.
>
> Thank you for your valuable comment.
>
> We would like to discuss why label smoothing works based on [1], and why assigning self-knowledge distillation to label smoothing is effective.
>
> The effectiveness of the label smoothing has been thoroughly discussed in [1]. To briefly summarize, the authors state that label smoothing encourages the last hidden representation (before classification layer) to be **equally distant to the embeddings of the incorrect classes (not target class)**, due to the uniform prior label distribution. This leads to why assigning knowledge distillation to label smoothing is effective. With prior label distribution being the "dark knowledge" from a teacher, the last hidden representation is encouraged to be closer to the incorrect classes that have high predictive scores by a teacher network. Therefore, an incorrect class with low predictive score by a teacher will be more distant to the hidden representation, while an incorrect class with high predictive score by a teacher will be closer to the hidden representation. In conclusion, the hidden representation will be more close to the "plausible" classes, determined by a teacher, than the less likely classes.
>
> [1] When Does Label Smoothing Help?, NIPS 2019
>
> ### Response 2
> > The authors claim that “There are a number of benefits of adopting the adaptive smoothing parameter”. However, they only show that the hyperparameter search on $\alpha$ is removed and the adaptive smoothing parameter can be connected to the gradient rescaling effect on self-distillation. More details should be reported to show the benefits of adopting the adaptive smoothing parameter.
>
> Thank you for the clear understanding of our paper. You are correct in the two core benefits of our adaptive smoothing parameter which lead to improvement in generalization ability and model calibration. We have changed the phrase "a number of" to "two" in the updated version. Thank you for pointing it out.
>
> ### Response 3
> > In the ablation study, the authors only consider the fixed $\alpha$ as the base. However, $\alpha$ could also be changed in the training process. Therefore, the dynamic $\alpha$ by hyperparameter searching should also be added as a base.
>
> Thank you for the helpful comment. Following your review, we have added the dynamic $\alpha$ in our ablation study, in which the alpha value is linearly increased throughout the course of training. The result is as follows:
> ### Linear Increase Alpha vs Our Adaptive Alpha on IWSLT DE->EN
> |Method|BLEU (&#8593;)|METEOR (&#8593;)|WER (&#8595;)| ROUGE-L (&#8593;)|NIST (&#8593;)|ECE (&#8595;)|MCE (&#8595;)|
> |:----------:|:-------------:|:------:|:------:|:------:|:------:|:------:|:------:|
> | Linear Increase $\alpha$ | 37.52| 65.92|46.54	| 63.04|8.72 |	5.58|	8.94 |
> | Our Adaptive $\alpha$ | **37.82**|**66.24**|**46.09**|**63.41**|**8.79**|	**1.76**|**3.64**|
>
> The result clearly illustrates the effectiveness of our adaptive alpha, especially in model calibration scores. This is related to the gradient rescaling factor brought by our adaptive alpha, so that the samples with a sign of overconfidence receive higher regularization effect. The result is now added to the modified version of our paper.
>
> For the hyperparameter search, we have tried {0.2, 0.4, 0.6, 0.8} for maximum $\alpha$ and {150, 200, 250, 300} for the maximum training epoch. We find that maximum alpha with 0.8 and maximum epoch 300 show the best evaluation performance, hence we report the score in the ablation table.
>
> ### Response 4
> Thank you for suggesting the papers. We have implemented and added LORAS [2] to one of our baselines. The changes are made in the updated version.
>
> [2] Learning Better Structured Representations Using Low-Rank Adaptive Label Smoothing, ICLR 2021

---

> > ### Comment · Reviewer_xcg8 · 2021-11-20
> > **Response**
> >
> > The main concerns are solved. Thank you.

---

> ### Author Response · Authors · 2021-11-20
> **Dear Reviewer xcg8**
>
> We hoped our response helped resolve your concerns regarding ablation and connection between KD and label smoothing. If you have any further comments/questions, we would be happy to answer them.

---

### Official Review · Reviewer_WMdY · 2021-10-30

**Correctness:** 4
**Technical Novelty And Significance:** 4
**Empirical Novelty And Significance:** 4
**Recommendation:** 6
**Confidence:** 4

**Main Review:**


1. Strengths

The paper proposes a simple extension for label smoothing that enables adaptive and instance-level smoothing. Experiments show good results w.r.t. model performance measures and calibration error metrics, even when compared with recent self-distillation methods such as TF-KD and SKD-PRT. Ablation study and gradient analysis provide further insights into the proposed method. It is also nice that the smoothing hyper-parameter tuning is not required anymore.

2. Weaknesses

There are a few instance-level label smoothing methods that are not discussed (nor compared) in the paper: Li et al. "Regularization via structural label smoothing" (AISTATS2020). Zhang and Sabuncu "Self-distillation as instance-specific label smoothing" (NeurIPS2020).

Although already discussed in the final section, it would be interesting to see results for datasets from other domains, such as images. Since most label smoothing methods were proposed to work well with image datasets, it would make the paper much better to show how it will perform under under other tasks/datasets, such as image classification.

**Summary Of The Paper:**

This paper proposes a new method that is based on label smoothing.
The original label smoothing uses the same level of smoothness among samples and among time steps.
This paper tries to extend this to a dynamic nature with different smoothness between samples and also throughout training.
The main idea is to use a normalized version of the entropic level of model prediction for a certain input data point for the smoothness parameter.
This is combined with self-knowledge which is used as the distribution of label smoothing.

**Summary Of The Review:**

This paper provides a simple way to extend label smoothing to instance-specific label smoothing with an adaptive smoothing parameter. The benefits of the method is shown empirically, with an ablation study, and through gradient analysis.

After rebuttal: Thank you for the additional experiments. It is good to see that the proposed method tend to perform better than the additional baselines. I do not have further comments/questions.

---

> ### Author Response · Authors · 2021-11-15
> **Response to Reviewer WMdY**
>
> ### Response 1
> > There are a few instance-level label smoothing methods that are not discussed (nor compared) in the paper: Li et al. "Regularization via structural label smoothing" (AISTATS2020). Zhang and Sabuncu "Self-distillation as instance-specific label smoothing" (NeurIPS2020).
>
> Thank you for your suggestions for the papers. We have added reference to the papers. We have implemented 3 more baseline methods that are domain-agnostic (including Beta Sampling and Self-Knowledge Distillation from [1]). The changes are made in the modified version.
>
> [1] Self-Distillation as Instance-Specific Label Smoothing, NIPS 2020
> ### Response 2
> > Although already discussed in the final section, it would be interesting to see results for datasets from other domains, such as images. Since most label smoothing methods were proposed to work well with image datasets, it would make the paper much better to show how it will perform under other tasks/datasets, such as image classification.
>
> Thank you for your insightful review. Our main focus of the evaluation was set to NLP, especially the Natural Language Generation (NLG) task, as it is a challenging task in terms of smoothing a label. In NLG, the number of classes is comparatively large (around 6.7K for IWSLT14 and IWSLT15 and 7K for Multi30K), making the task challenging, in addition to the challenges brought by the autoregressive aspect of a language model. We would like to emphasize that our method is task and domain-agnostic, and it can easily be extended to other classification tasks. We hope to see our method being tested in other tasks/domains in the future.

---

> ### Author Response · Authors · 2021-11-20
> **Dear Reviewer WMdY**
>
> If you have any further questions/comments, we would be happy to address them.

---

### Official Review · Reviewer_528k · 2021-10-31

**Correctness:** 4
**Technical Novelty And Significance:** 2
**Empirical Novelty And Significance:** 2
**Recommendation:** 6
**Confidence:** 4

**Main Review:**

Strengths
- Overall, the paper is well written, and the proposed method is well motivated.
- Experiments seem to demonstrate the effectiveness of the proposed method.
- Thorough empirical analysis is done on several benchmark datasets.
- A careful ablation study is also conducted to further demonstrate the effectiveness of the proposed method.

Weakness
- The main weakness of the paper in my opinion is the lack of novelty. The proposed method in my opinion is very similar to the previous works like [1], which proposed to use predictions from previous time stamps for self-distillation, and [2], which also proposed a method for adaptive label smoothing based on predictions from previous time stamps. While the authors of the paper addressed the key difference between the proposed method and [1], I do still feel that the proposed method lack novelty, despite good improvements in performance. On top of [1] and [2], [3] and [4] are also potentially relevant prio works to potentially discuss in literature review and benchmark against.

[1] Kyungyul Kim, ByeongMoon Ji, Doyoung Yoon, and Sangheum Hwang. Self-knowledge distillation with progressive refinement of targets, 2021.

[2] Zhang, Zhilu, and Mert R. Sabuncu. "Self-distillation as instance-specific label smoothing." arXiv preprint arXiv:2006.05065 (2020).

[3] Li, Xingjian, et al. "One Generation Knowledge Distillation by Utilizing Peer Samples." (2019).

[4] Yun, Sukmin, et al. "Regularizing Predictions via Class-wise Self-knowledge Distillation." (2019).


**Summary Of The Paper:**

Label smoothing is a popular approach to regularize modern neural networks. However, the amount of smoothing is the same across all samples in the dataset, which can be sub-optimal. In this paper, the authors of the paper proposed a novel adaptive label smoothing method so that each sample gets a different amount of smoothing. The key insight of the paper involves using the entropy of predictions from past timestamps as a way to quantify the amount of label smoothing applied to samples. The authors of the paper demonstrate empirically the effectiveness of the proposed method.


**Summary Of The Review:**

Overall, despite the good empirical performance, I think the lack of novelty is a significant weakness of the paper. As such, I recommend weakly rejecting the paper for now.

After Rebuttal:
I would like to thank the authors of the paper for all clarifications and additional experiments. After reading the response and other reviewers' comments, I am raising my score to 6.

---

> ### Author Response · Authors · 2021-11-15
> **Response to Reviewer 528k (1/2)**
>
> Thank you for your insightful review on our work. The following are the responses to the points made in your comment.
>
> ### Response 1
> > The proposed method in my opinion is very similar to the previous works like [1], which proposed to use predictions from previous time stamps for self-distillation
>
> First, we would like to make clear that our work significantly differs from SKD-PRT [1] in several ways.
>
> 1. Smoothing Parameter ($\alpha$)
>
> The first core difference is adaptive smoothing parameter ($\alpha$). SKD-PRT [1] employs a scheduled alpha scheme, in which the alpha value is gradually increased throughout the training, hence the alpha is identical in each sample and each time-step. In our work, the alpha differs in every time-step and every sample. This aspect of ours enables gradient resailing with respect to alpha, while there is no gradient resailing effect by alpha in [1].  This core difference is shown with empirical evaluation in terms of translation quality and model calibration and with gradient analysis.
>
> Aside from the theoretical analysis, we added one more ablation study to empirically show the effectiveness of our adaptive $\alpha$ compared to the scheduled alpha. We have added an ablation study with linear increase in smoothing parameter (as in [1]), and compared the results with our version of smoothing parameter. The following is the result.
> ### Linear Increase Alpha vs Our Adaptive Alpha on IWSLT DE->EN
> |Method|BLEU (&#8593;)|METEOR (&#8593;)|WER (&#8595;)| ROUGE-L (&#8593;)|NIST (&#8593;)|ECE (&#8595;)|MCE (&#8595;)|
> |:----------:|:-------------:|:------:|:------:|:------:|:------:|:------:|:------:|
> | Linear Increase $\alpha$ | 37.52| 65.92|46.54	| 63.04|8.72 |	5.58|	8.94 |
> | Our Adaptive $\alpha$ | **37.82**|**66.24**|**46.09**|**63.41**|**8.79**|	**1.76**|**3.64**|
>
> This table clearly shows the effectiveness of our adaptive alpha. The most **noticable difference comes from ECE and MCE score**, as expected. The two scores on model calibration are noticeably higher than those of our adaptive alpha, and this is due to the gradient rescaling factor that comes from our adaptive alpha.
>
> 2. Gradient Analysis
>
> Another difference comes from the gradient analysis. In Section 3 Theoretical Support in [1], the authors discuss the gradient analysis in terms of **L1 norm**. However, **L1 norm cannot depict the change of *direction* of the gradient.** In our work, we discuss the importance of alpha, such that high alpha is more likely to cause change in *direction of the gradient*, compared to that of the cross-entropy.
>
> In addition, the interpretation of the gradient analysis differs. [1] argues that the gradient rescaling factor is greater on "hard-to-learn" samples than "easy-to-learn" samples. We further extend the argument in terms of the adaptive alpha, and argue that the regularization effect is greater on the samples with signs of overconfidence. This difference is brought with the introduction of adaptive alpha.
>
> 3. Function $g$
>
> Another difference from [1] is how our model identifies the self-teacher checkpoint. [1] sets epoch t-1 as a teacher network. However, our work utilizes the function $g$ for selecting which checkpoint among the past checkpoints to choose.
>
> 4. $\alpha$-specific Hyperparameter Search
>
> [1] utilizes a linear increase scheme on alpha. This, however, involves two hyperparameters: max alpha, and max epoch. The first hyperparameter has a fixed search window between 0 and 1, yet the second does not; This brings high search space in finding the optimal hyperparameters. Our work removes the need for regularization-specific hyperparameter search, as the smoothing parameter is computed on the fly during the forward propagation. The self-regulation aspect in the alpha is shown with Figure 2(a).
>
> [1] Self-Knowledge Distillation with Progressive Refinement of Targets, ICCV 2021

---

> ### Author Response · Authors · 2021-11-15
> **Response to Reviewer 528k (2/2)**
>
> ### Response 2
> > The main weakness of the paper in my opinion is the lack of novelty. The proposed method in my opinion is very similar to the previous works like [1], which proposed to use predictions from previous time stamps for self-distillation, and [2], which also proposed a method for adaptive label smoothing based on predictions from previous time stamps.
>
>
> We would like to clarify the novelty of the paper. Our paper falls into the Self-Knowledge Distillation category, in which researchers utilize past or pre-trained checkpoint of the identical model structure. Therefore, utilizing self-knowledge is not the main contribution of our paper. The core novelty of our work is how we **adaptively balance** the ground-truth and prior label distribution (Self-Knowledge) in the aim of improving generalization and model calibration. Previous methods focus on how to come up with a meaningful prior label distribution. Our main contribution, on top of finding a prior label distribution, is how to effectively and adaptively incorporate the prior label distribution with the ground-truth. For example, [2] propose Beta smoothing and Self-Distillation. Their work is closely related on how to come up with the prior label distribution for each instance, **yet with fixed alpha**. Therefore, their **"adaptive" applies to prior label distribution**, and the extent of smoothing remains fixed. However, our **"adaptive" refers to both 1) adaptive prior label distribution, and 2) adaptive alpha.**. The amount of smoothing is instance-specific and the prior distribution is also instance-specific in our work. Therefore, gradient rescaling by alpha is only possible with our method.
>
> In addition, we make meaningful contributions with the gradient analysis related to alpha, and show how changes in the direction of the gradient can be made. Our work not only discusses the theoretical analysis, but also illustrates the effectiveness of the proposed method with empirical results. To the best of our knowledge, there is no prior work that directly utilizes the model probability distribution as a balancing parameter between ground-truth and self-knowledge, and discusses why the dynamic nature of alpha is crucial in improving model generatlization and calibration.
>
> We hope that your concern regarding the novelty of our work has been resolved.
>
> [2] Self-Distillation as Instance-Specific Label Smoothing, NIPS 2020 ([Link to Official Code Repository](https://github.com/ZhiluZhang123/neurips_2020_distillation.git))
>
> ### Response 3
> > On top of [1] and [2], [3] and [4] are also potentially relevant prior works to potentially discuss in literature review and benchmark against.
>
> Thank you for your suggestions. We have cited and discussed the papers, and also we have implemented 3 more baseline methods that are domain-agnostic (including 1) SelfDistillation and 2) Beta Smoothing from [2]). The changes are made in the modified version.

---

> ### Author Response · Authors · 2021-11-20
> **Dear Reviewer 528k**
>
> We would like to thank you for understanding the novelty of our paper. If you have any further questions/comments, we are more than happy to address them.

---

### Author Response · Authors · 2021-11-15
**General Comment**

We would like to first thank the reviewers for their thorough and insightful comments.

As some of the reviewers recommended to discuss/compare previous work related, we have cited and discussed related papers mentioned in the reviewer's comments. Furthermore, we have added 3 domain-agnostic baseline methods (SD, BETA from [1], and LORAS from [2]), hence the paper now evaluates 9 baseline methods.

In addition, to further validate the effectiveness of the proposed method, we have included two more ablation studies. 1) Linearly increasing alpha and 2) Different $g$ function.

Summary of the major changes made in the updated version
1. 3 more methods are added for evaluation (SD, BETA from [1] and LORAS from [2], total of 9 baselines now)
2. 2 more ablation studies (linear increase in alpha and different $g$ function)
3. Added references of the previous papers that were mentioned by the reviewers

The following are the results of the 3 newly added baselines. The full table along with other baselines can be found in the paper.
### **Multi30K** - Automatic Evaluation Result (Comparison with the **newly added baselines**)
|Method|BLEU (&#8593;)|METEOR (&#8593;)|WER (&#8595;)| ROUGE-L (&#8593;)|NIST (&#8593;)|
|:-:|:----------:|:-------------:|:------:|:------:|:------:|
|Base| 40.64| 73.31|38.76| 69.21| 7.96|
|LORAS [2]| +1.14	|+1.00	|-0.73	|+0.63	|+0.16|
|BETA [1]| +1.26	|+0.94 |-0.20	|+0.46	|+0.07 |
|SD [1]| +2.76	|+2.09	|-1.26 |+1.55	|+0.18|
|Ours | **+3.75** | **+2.91**| **-2.19**| **+2.17**| **+0.32**|

### **IWSLT15** - Automatic Evaluation Result (Comparison with the **newly added baselines**)
|Method|BLEU (&#8593;)|METEOR (&#8593;)|WER (&#8595;)| ROUGE-L (&#8593;)|NIST (&#8593;)|
|:-:|:----------:|:-------------:|:------:|:------:|:------:|
|Base| 30.17 | 59.09 | 54.02 | 63.91| 7.18|
|LORAS [2]|-0.04| -0.10|-0.23|-0.19|+0.02|
|BETA [1]|+0.30|+0.20|-0.63|+0.17|+0.07|
|SD [1]|+0.68|+0.46|-0.63|+0.45|+0.08|
|Ours|**+1.37**|**+1.14**|**-1.80**|**+1.05**|**+0.21**|

### **IWSLT14** - Automatic Evaluation Result (Comparison with the **newly added baselines**)
|Method|BLEU (&#8593;)|METEOR (&#8593;)|WER (&#8595;)| ROUGE-L (&#8593;)|NIST (&#8593;)|
|:-:|:----------:|:-------------:|:------:|:------:|:------:|
|Base|35.96|64.70|48.17|61.82|8.47|
|LORAS [2]|+0.36|+0.23|+0.61|+0.16|-0.03|
|BETA [1]|+0.95|+0.69|-0.30|+0.58|+0.08|
|SD [1]|+1.39|+1.06|-0.84|+0.88|+0.16|
|Ours|**+1.86**|**+1.55**|**-2.08**|**+1.59**|**+0.32**|

### **Model Calibration** Related Result (Comparison with the **newly added baselines**)
|Method|Multi30K ECE|Multi30K MCE|IWSLT15 ECE|IWSLT15 MCE|IWSLT14 ECE|IWSLT14 MCE|
|:-:|:-:|:-:|:-:|:-:|:-:|:-:|
|Base|14.95| 26.01| 14.05| 20.38 | 12.98|19.29|
|LORAS [2]|20.27 | 40.86| 12.41| 19.15|10.54 |15.29|
|BETA [1]| 11.71 | 21.90 | 9.57|14.97|8.63 |13.21|
|SD [1]| 6.87 | **12.38**| 5.01|9.64|7.82 |13.71|
|Ours | **4.76** | 12.41| **2.15**| **4.40**|**1.76**| **3.64**|


[1] Self-Distillation as Instance-Specific Label Smoothing, NIPS 2020

[2] Learning Better Structured Representations Using Low-Rank Adaptive Label Smoothing, ICLR 2021

---

### Decision · Program_Chairs · 2022-01-20

**Decision:**

Reject

**Comment:**

The paper proposes an approach to performing label smoothing, with the amount of smoothing being sample-dependent and guided by the model's prediction (similar to self-distillation). While the reviewers find the studied problem relevant and important, they find the contributions (in their current state) to be borderline, mainly on the basis of lack of novelty and missing discussion with some related papers. While authors' response was able to partially resolve these concerns, at the end none of the reviewers was a strong advocate for accepting the paper and all scores remained at the borderline (although on the positive side). In concordance with the reviewers, I believe this submission can be made much stronger by digging a bit deeper into the problem, and also making broader connections with the existing literature.

As a concrete example/suggestion (among many other possibilities for strengthening this work), the authors may want to go a bit deeper into the theoretical analysis. Currently, their analysis shows the approach is able to reduce model's confidence, which is what happens in label smoothing and self-distillation. However, self-distillation is more than confidence reduction, and the information contained in the "dark knowledge" can provide a much stronger regularization than a sole confidence reduction argument. There are already some papers in the literature on the regularization/generalization effects of self-distillation, which the authors might want to use as a stepping-stone.